# The role of *OLE2* and *POX1-3* in prostaglandin E$_2$ production and virulence is conserved in *Candidozyma* (*Candida*) *auris*

Armand Bolsenbroek,[1] Eduvan Bisschoff,[1] Gabre Kemp,[1] Olihile Sebolai,[1] Carolina Pohl,[1] Jacobus Albertyn[1]

**ABSTRACT** Lipids have recently gained attention for their involvement in the virulence of pathogenic yeast. One group of lipids, prostaglandins, which are arachidonic acid-derived eicosanoids, has received particular attention. Prostaglandin E$_2$ (PGE$_2$) is the most studied prostaglandin in mammals. However, pathogenic yeast, including *Candida albicans*, *Candida tropicalis*, *Candida glabrata*, *Candida parapsilosis*, and *Candida dubliniensis*, also produces PGE$_2$. Notably, PGE$_2$ production has not yet been reported in *Candidozyma auris* (*Candida auris*). In *C. albicans*, PGE$_2$ plays a crucial role in colonization and virulence by promoting yeast-to-hyphal transition, enhancing biofilm formation and colonization of the gastrointestinal tract. Although the biochemical pathway responsible for PGE$_2$ synthesis in yeasts is unknown, several genes have been implicated, including *OLE2* and *POX1-3*, which were investigated in this study. Deletion mutants of these genes were generated using a CRISPR-Cas9 system, and the production of PGE$_2$ by the mutants was quantified using enzyme-linked immunosorbent assay and confirmed using liquid chromatography-tandem mass spectrometry. Furthermore, the involvement of these genes in virulence was assessed using a survival assay in *Caenorhabditis elegans*. This study presents the first report of PGE$_2$ production by *C. auris*, and it demonstrates that *OLE2* and *POX1-3* play significant roles in PGE$_2$ production and virulence of this yeast in *C. elegans*. Additionally, deletion of *OLE2* in *C. auris* led to an accumulation of monounsaturated fatty acids, providing evidence that *OLE2* does not encode a Δ9 desaturase. The conserved nature of the PGE$_2$ production and virulence requirements among different pathogenic yeasts indicates the potential for broad-spectrum drug targeting.

**IMPORTANCE** This study reveals, for the first time, that *Candidozyma* (*Candida auris*)—a growing global health threat, often causing outbreaks in hospitals—can produce a prostaglandin E$_2$ (PGE$_2$), which is known to influence how the immune system responds and has been linked to increased virulence in other yeast species. By identifying specific genes (*OLE2* and *POX1-3*) involved in PGE$_2$ production and virulence, potential novel drug targets have been identified. Understanding how *C. auris* produces PGE2 and how this contributes to its ability to infect and survive in hosts could lead to innovative therapies that block these pathways, making infections easier to treat.

**KEYWORDS** pathogenic yeast, *Candidozyma auris (Candida auris)*, CRISPR-Cas9, prostaglandin E$_2$, immunomodulatory lipids

Eicosanoids derived from polyunsaturated fatty acids (PUFAs) such as arachidonic acid (AA), eicosapentaenoic acid, and docosahexaenoic acid include prostaglandins, thromboxanes, prostacyclins, and leukotrienes (1, 2) and represent a diverse and essential class of signaling lipids involved in a wide range of physiological and pathophysiological processes. The biosynthesis of prostaglandins in mammalian cells is a well-established process in which AA, a PUFA present in the cell membrane, is liberated by phospholipases and subsequently undergoes conversion by cyclooxygenase (COX)

**Peer Reviewer** Feng Yang, Tenth People's Hospital of Tongji University, Shanghai, China

Address correspondence to Carolina Pohl, PohlCH@ufs.ac.za, or Jacobus Albertyn, AlbertynJ@ufs.ac.za.

The authors declare no conflict of interest.

See the funding table on p. 9.

enzymes to generate prostaglandin $H_2$ ($PGH_2$). $PGH_2$ serves as a common precursor for synthesizing diverse prostaglandins belonging to different classes, such as prostaglandin D, F, I, and E (1, 2). The expression and activity of distinct downstream enzymes, including prostaglandin synthases and isomerases, determine the ultimate production of specific prostaglandins.

The best studied of these eicosanoids is prostaglandin $E_2$ ($PGE_2$), which exhibits immunomodulatory effects in mammals in response to various stimuli, including pathogens (3). It can attenuate dendritic cell maturation and modulate the phagocytic activity of monocytes, macrophages, and neutrophils by selectively inhibiting their function. Additionally, $PGE_2$ can suppress the Th1 inflammatory response, essential for host defense against *Candida albicans* infection, while promoting the Th2 and Th17 inflammatory responses (4–8).

In addition to mammals, eicosanoids can be biosynthesized by yeast and have been observed in pathogenic yeasts such as *C. albicans*. The first eicosanoid identified to be produced by *C. albicans* from exogenous AA was 3,18-dihydroxy-5,8,11,14-eicosatetraenoic acid (9). The production of $PGE_2$ by *C. albicans* was also confirmed (10–12) and subsequently verified in several other *Candida* spp., such as *Candida dubliniensis*, *Candida parapsilosis*, *Candida glabrata*, and *Candida tropicalis* (13–16).

Interestingly, whole-genome sequencing indicated the absence of shared enzymatic pathways for prostaglandin synthesis between mammals and *Candida* spp. (11, 12, 17). In addition, the selective COX-2 inhibitor, CAY10404 (3-[4-methylsulfonylphenyl]–4-phenyl-5-trifluoromethylisoxazole), did not decrease $PGE_2$ production in *C. albicans*, indicating that these yeasts utilize a distinct pathway for $PGE_2$ synthesis (11). Further investigation revealed that the enzymes that play a role in $PGE_2$ production in *Candida* spp. include a putative stearyl coenzyme A desaturase (encoded by *OLE2*), which is implicated in *C. albicans*. Homozygous deletion mutants lacking *OLE2* show significantly decreased $PGE_2$ production (11), as did deletion of genes of the high-affinity reductive iron acquisition pathway, such as *FET99* (encoding a multicopper oxidase involved in iron transport) (18). Interestingly, in the case of *C. parapsilosis,* $PGE_2$ production was not affected by deletion of *OLE2* but was negatively influenced by deletion of *FET3* (encoding a multicopper oxidase involved in iron transport), *POX1-3* (encoding an acyl-CoA oxidase), and *POT1* (encoding an acyl-CoA thiolase). This suggests that although there are similarities regarding the enzymes that influence $PGE_2$ production in different *Candida* spp., there are also differences between them.

*Candida auris*, which was first identified in 2009 (19), is a newly emerged pathogenic yeast that poses a severe threat to hospital patients worldwide. It is highly persistent in healthcare facilities, causing multiple nosocomial outbreaks, and the rapid development of antifungal resistance limits the treatment options (20). This yeast was the first fungal pathogen categorized as a public health threat by the Centers for Disease Control and Prevention, and the World Health Organization has recently included *C. auris* in the highest group (critical) of priority fungal pathogens with *C. albicans*, *Cryptococcus neoformans*, and *Aspergillus fumigatus* (21). Thus, it is important to gain a better understanding of the biology of this yeast, including pathways involved in $PGE_2$ production, which may serve as novel antifungal drug targets (22).

Therefore, this study aims to investigate the production of $PGE_2$ in *C. auris*. Specifically, we assessed the involvement of *OLE2* and *POX1-3,* previously linked to $PGE_2$ production in certain *Candida* spp. Furthermore, we explored the potential contribution of these genes to *C. auris* virulence.

## MATERIALS AND METHODS

### Strains used

The strains that were used are listed in Table 1. *C. albicans* SC5314 was a kind gift of Prof. B. Hube (Leibniz Institute for Natural Product Research and Infection Biology—Hans Knöll Institute, Germany), and *C. auris* MRU293 (= B11224) was provided by Dr.

**TABLE 1** Strains used in this study

| Species | Genotype | Parental strain | Description | Origin |
|---------|----------|-----------------|-------------|--------|
| *C. albicans* SC5314 | Wild type | –[a] | Reference strain | N/A[b] |
| *C. auris* MRU293 | Wild type | – | Clinical strain | N/A |
| *C. auris* | *ole2Δ* | MRU293 | Deletion of *OLE2* | This study |
| *C. auris* | *ole2Δ::OLE2* | MRU293 | Add-back of *OLE2* | This study |
| *C. auris* | *pox1-3Δ* | MRU293 | Deletion of *POX1-3* | This study |
| *C. auris* | *pox1-3Δ::POX1-3* | MRU293 | Add-back of *POX1-3* | This study |

[a]"–" indicates no parental strain.
[b]N/A, not applicable.

Nelesh Govender (National Institute for Communicable Diseases, South Africa). *C. auris* strains and mutants were stored at −80°C in 30% glycerol and revived on yeast peptone dextrose (YPD; 10 g/L yeast extract, 20 g/L peptone, and 20 g/L glucose) at 30°C.

## Construction of *C. auris* mutants

A published CRISPR-Cas9 system (23) for *C. albicans* was adapted for use in *C. auris* MRU293 by replacing *C. albicans*-specific sequences with *C. auris* sequences. This system introduces a double-strand break at the target gene, and then modification occurs through the donor DNA in the wild type. A detailed explanation of the methods to obtain these mutants can be found in the supplemental material. After deletion mutants (*ole2Δ* and *pox1-3Δ*) were generated, add-back strains were generated by reintroduction of the wild-type gene by modified donor DNA.

## Biofilm formation and biomass determination

All strains were streaked onto yeast malt (YM) extract medium agar (3 g/L malt extract, 3 g/L yeast extract, 5 g/L peptone, 10 g/L glucose, and 16 g/L agar) and incubated for 24 h at 30°C. Single colonies on the YM plates were inoculated into 5 mL of yeast nitrogen base (YNB) broth (16 g/L YNB and 10 g/L glucose) and incubated at 30°C for 24 h with shaking. The cells were harvested by centrifugation at 1,878 × *g* for 5 min, and the supernatant was discarded. The cells were washed three times with sterile phosphate-buffered saline (Oxford, UK) and standardized to $1 \times 10^6$ cells/mL in 20 mL filter-sterilized (0.20 µm polyethylene syringe filter, GVS Life Sciences, USA) RPMI-1640 medium (Sigma-Aldrich, USA). Biofilms were grown in 90 mm polystyrene Petri dishes in the presence of 500 µM AA (Sigma-Aldrich, USA), for 48 h at 37°C. After incubation, the biofilm was scraped off and filtered through pre-weighed filters (0.22 µm nitrocellulose filter, Sartorius, Germany), and the supernatant was collected in 50 mL conical tubes. The supernatants were stored at −80°C until further use. The nitrocellulose filter was dried at 37°C for 48 h and weighed to calculate the biomass.

## Liquid chromatography-tandem mass spectrometry

LC-MS/MS analysis was conducted to confirm the production of PGE$_2$ by *C. auris*. An API3200QTRAP hybrid triple quadrupole linear ion trap mass spectrometer (ABSciex, Canada) was coupled with an Agilent 1,200 SL series high-performance liquid chromatography (HPLC) system. The samples (10 µL) were injected at a flow rate of 0.5 mL/min, and the analysis was performed at 30°C. The samples were separated using a Zorbax Eclipse XDB C18, 50 × 4.6 mm column (Agilent Technologies, Germany). The mobile phases employed were solvent A and solvent B. Solvent A was a 10 mM ammonium formate aqueous solution in 5% methanol, while solvent B was a 10 mM ammonium formate aqueous solution in 95% methanol. The elution gradient consisted of 0–5 min: 50% B, 5–10 min: 50%–95% B, followed by an equilibration step, resulting in a total chromatographic run of 20 min. Atmospheric pressure electrospray ionization was performed in the negative mode. PGE$_2$ reference standard (Cayman Chemicals, USA) was included in the compound optimization feature in the Analyst software

to develop a multiple reaction monitoring (MRM) method with five transitions (one precursor producing five unique fragments) prior to sample separation. The ion spray voltage was set at 4,500 V, and the source temperature was maintained at 500℃. The declustering potential was 20 V, and the collision energy ranged from 14 to 30 eV for the fragment ions. The five selected transitions were 351.17/315.2, 351.17/271.2, 351.17/333.3, 351.17/189, and 351.17/235.1. Only when all five transitions were detected at the same retention time was the presence of $PGE_2$ in the sample confirmed.

## Quantification of $PGE_2$ production by *C. auris*

Supernatants collected from the biofilms were acidified with 1M formic acid to a pH of less than 4. Lipids were extracted using solid-phase extraction (SPE) C18-T cartridges (Phenomenex, USA). The SPE cartridges were washed with 5 mL of methanol (Merck, Germany) and 5 mL of deionized water. After column preparation, 10 mL of acidified supernatant was added and allowed to flow through the column. Subsequently, 5 mL of water was used to remove the impurities. The lipids were eluted using 5 mL of ethyl acetate containing 1% methanol and collected in conical tubes wrapped in foil. The eluent was then dried under a stream of $N_2$ gas, and the $PGE_2$ concentration was determined using ELISA (Caymen Chemicals, USA) according to the manufacturer's instructions. A cell-free control containing AA was included, and the background values obtained were subtracted from the experimental values. The experiment was performed in duplicate, with each sample assayed at two different dilutions, which were assayed in triplicate. The data were analyzed according to the manufacturer's specifications, and the concentrations were normalized against the biofilm biomass (Fig. S1 and S2).

## Determination of lipid profile of *C. auris* strains

Biofilms were prepared as described above for each strain. The cells were harvested, stored at −80℃ overnight, and then freeze-dried. The cells were then suspended in a 2:1 mixture of chloroform:methanol, left overnight, filtered, and dried using a rotary evaporator. The dried filtrates (containing extracted lipids) were dissolved in chloroform, and the fatty acid was transesterified with trimethylsulfonium hydroxide (24). Fatty acid methyl esters were subsequently analyzed on a Shimadzu GC-2010 gas chromatograph with a flame ionization detector and autosampler. Injection volume was 0.5 µL, the injection port temperature was 275℃, and the split ratio was 1:10. An SGE-BPX90 column (length: 60 m, inner diameter: 0.25 mm, and film thickness 0.25 µm) was used. Hydrogen was the carrier gas at 60 cm/s linear velocity (3.19 mL/min). Initial oven temperature was 80℃, held for 1 minute, and then ramped at 10℃–280℃ and held for 4 minutes. Instrument control, data collection, and analysis were performed with Shimadzu LabSolutions software. The identity of the analytes was confirmed by comparison of retention times with that of authentic standards.

## *Caenorhabditis elegans* survival assay

The *Caenorhabditis elegans glp-4; sek-1* strain was used in this study and obtained from the Caenorhabditis Genetic Center, College of Biological Science, University of Minnesota. Stocks were kept at 15℃ on nematode growth medium (NGM; 2.5 g/L peptone; 15 g/L agar; and 3 g/L sodium chloride). Synchronized L4 nematodes were washed off NGM plates using M9 buffer (6 g/L $Na_2HPO_4$; 5 g/L NaCl; 3 g/L $KH_2PO_4$; and 0.25 g/L $MgSO_4$) into 50 mL conical tubes and washed four times with M9 buffer. Following the wash, the nematodes were infected by spreading them on brain-heart infusion (BHI; Sigma-Aldrich) agar plates (7.8 g/L brain extract; 9.7 g/L heart extract; 2.0 g/L dextrose; 2.5 g/L disodium phosphate; and 15 g/L agar) with *C. auris* and *C. auris* mutant lawns prepared as follows: The yeast strains were inoculated into 5 mL YPD (10 g/L yeast extract, 20 g/L peptone, and 20 g/L glucose), incubated overnight with shaking at 37℃, and standardized ($OD_{600}$ of 0.8). The standardized culture was plated onto BHI plates and incubated at 37℃ for 24 h to form a lawn. The nematodes were incubated for 4 h at 25℃. After incubation, the nematodes were washed off the BHI plates with M9 buffer in 50 mL conical tubes

and washed four times with M9 buffer. Approximately 60 nematodes were counted and transferred to a single well of a six-well plate containing liquid medium (20% BHI, 80% M9 buffer, and 90 µg/mL kanamycin). The nematodes were monitored for 7 days and counted each day as either alive or dead. They were considered dead if no movement was observed after mechanical stimulation (25).

## Statistical analysis

*C. elegans* survival was assessed using the Kaplan-Meier method, and differences were determined with the log-rank test using OASIS 2 with statistical analyses performed using two-way analysis of variance with Bonferroni correction (26). All the other experiments were performed in triplicate, and the average and standard deviation were calculated unless stated otherwise. The student *t*-test with Bonferroni correction was carried out to determine statistically significant differences between data sets. A *P*-value of ≤0.05 was considered significant.

## RESULTS

### *C. auris* produces PGE$_2$, and deletion of either *OLE2* or *POX1-3* decreases its production

The LC-MS/MS analyses confirmed the production of PGE$_2$ by *C. auris* MRU293 from exogenous AA (Fig. 1) as all five required transitions found in the standard are also

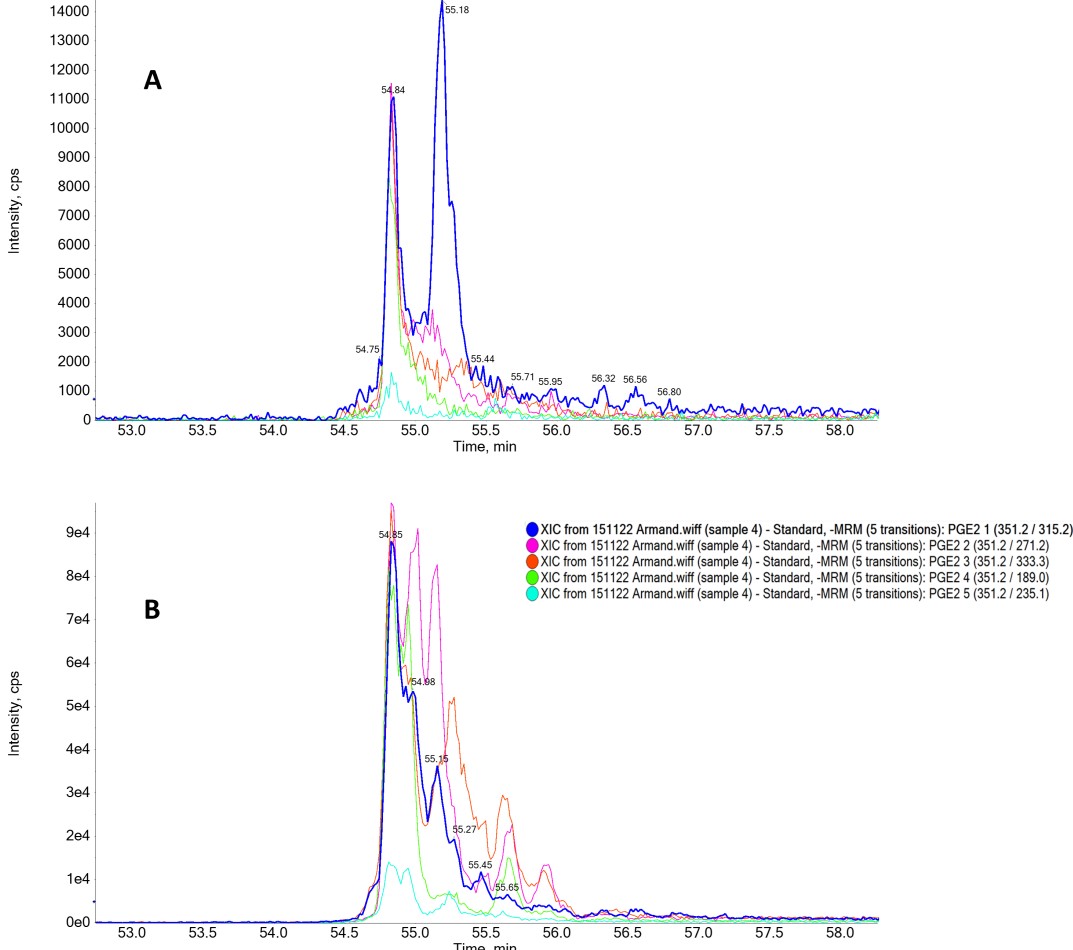

**FIG 1**  Chromatogram obtained with mass spectrometry displaying five transitions MRM spectra for two samples: (A) PGE$_2$ standard and (B) PGE$_2$ purified from *C. auris* biofilm cultivated in the presence of 500 µM AA. The relevant transitions are 351.2/315.2, 351.1/271.2, 351.2/333.3, 351.2/189.0, and 351.2/235.1.

present in the extract of *C. auris* at similar retention times. The role of *OLE2* and *POX1-3* in the production of PGE$_2$ by *C. auris* was determined by quantifying the PGE$_2$ concentrations in the supernatants of deletion mutants of these genes. *C. albicans* SC5314 was used as a reference strain for PGE$_2$ production (15). From Fig. 2, *C. albicans* SC5314 produced more PGE$_2$ than *C. auris* MRU293. Interestingly, *OLE2* and *POX1-3* are required for production of wild-type levels of PGE$_2$ by *C. auris*, since their mutants demonstrated decreased PGE$_2$ production, which was restored to wild-type levels in the add-back strains (*ole2Δ::OLE2* and *pox1-3Δ::POX1-3*).

## Deletion of *OLE2* causes an increase in the relative percentage of monounsaturated fatty acids

From Fig. 3, it is evident that deletion of *OLE2* or *POX1-3* impacts fatty acid profiles of *C. auris*. This is especially true for *OLE2*, where deletion caused a shift toward an increased relative percentage of palmitoleic acid (16:1) and oleic acid (18:1) at the expense of their precursors, palmitic acid (16:0) and stearic acid (18:0). This increase was reversed in the add-back strain. In addition, no change in the relative percentage of linolenic acid (18:2) and a decrease in the relative percentage of linolenic acid (18:3) were observed (Fig. 3). The effect of deletion of *POX1-3* was less clear as changes caused by the deletion mutant were not restored to wild-type levels in the add-back strains.

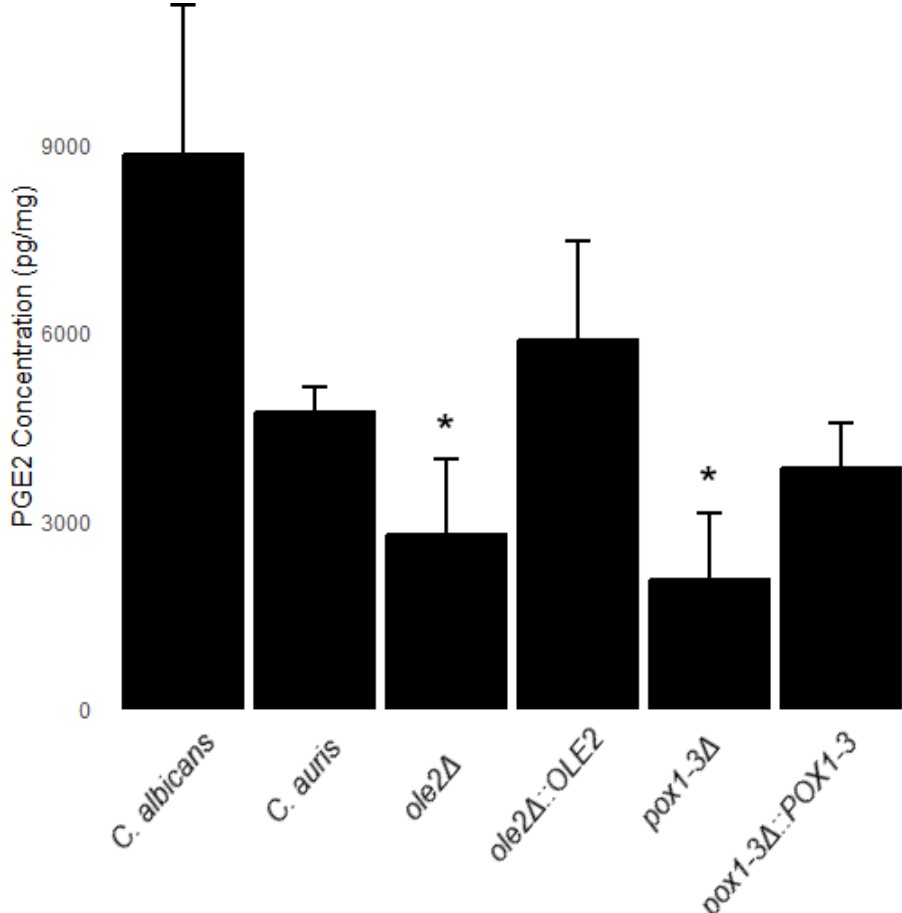

**FIG 2** PGE$_2$ production by *C. albicans* SC5314, *C. auris* MRU 293, and the mutant strains, following biofilm formation in the presence of 500 µM AA, as measured by ELISA. The analysis was conducted in triplicate, with each sample assayed at two different dilutions, which were assayed in triplicate, and the reported values represent the mean with the standard deviation indicated by the error bars. *Significantly different from *C. auris* MRU293 (*ole2Δ* $P = 0.029$; *pox1-3Δ* $P = 0.002$).

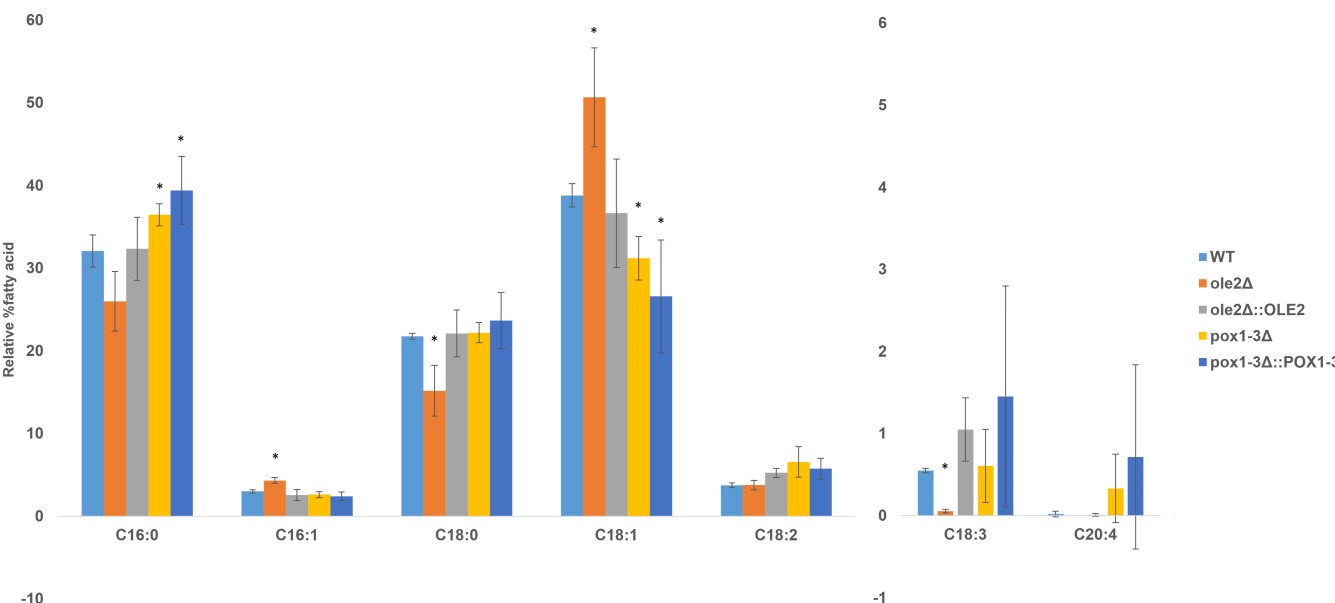

**FIG 3** Fatty acid profile of *C. auris* MRU 293 and the mutant strains, analyzed by gas chromatography. Values are the average of three biological repeats, and the standard deviation is indicated by the error bars. *Significantly different from the wild type ($P < 0.05$).

## *OLE2* and *POX1-3* are required for full virulence in *C. elegans*

To evaluate the role of genes implicated in PGE$_2$ production in virulence, *C. elegans* was infected with the different *C. auris* strains. The results were plotted in a Kaplan-Meier graph (Fig. 4), and data were analyzed (Table 2). The results showed a significant increase in the survival rates of the nematodes infected with the deletion mutants compared with those infected with the wild-type *C. auris*. Additionally, the add-back strains showed similar results to the wild type, thereby indicating the restoration of the function of the deleted mutants. These findings suggest that these genes have roles in the virulence of *C. auris* in this infection model.

## DISCUSSION

The synthesis of the immunomodulatory eicosanoid, PGE$_2$, has been established in various pathogenic *Candida* species, including *C. albicans*, *C. dubliniensis*, *C. parapsilosis*, *C. glabrata*, and *C. tropicalis* (13–16). The current work expands the conserved nature of the ability of yeasts to produce this immunomodulatory lipid molecule to *C. auris*. Although the metabolic pathway responsible for PGE$_2$ synthesis in these yeasts has not yet been elucidated, multiple genes have been implicated in PGE$_2$ synthesis (22). In this study, two genes, *OLE2* and *POX1-3*, were investigated and found to be involved in PGE$_2$ synthesis in *C. auris*.

*OLE2* encodes a putative fatty acid desaturase that may introduce double bonds into fatty acids (27). However, the deletion of *OLE2* in *C. albicans* did not influence fatty acid profiles of the cells, although overexpression of *OLE2* did cause a slight increase in 18:1 but did not influence the levels of other PUFAs. In *C. parapsilosis*, the deletion of *OLE2* caused an accumulation of 16:1 and 18:1, indicating that it does not encode a Δ9 desaturase (16). Similarly, we found that the deletion of *OLE2* in *C. auris* causes an accumulation of 16:1 and 18:1 at the expense of 16:0 and 18:0, providing further indication that this gene does not encode a Δ9 desaturase. It may be speculated that Ole2 could be involved in the downstream modification or turnover of monounsaturated fatty acids. However, although no change in the relative percentage of 18:2 was observed, there was a decrease in the relative percentage of 18:3, which may also indicate that Ole2 plays a role in the activity of the Δ15 desaturase, Fad3 (Fig. 3). Future

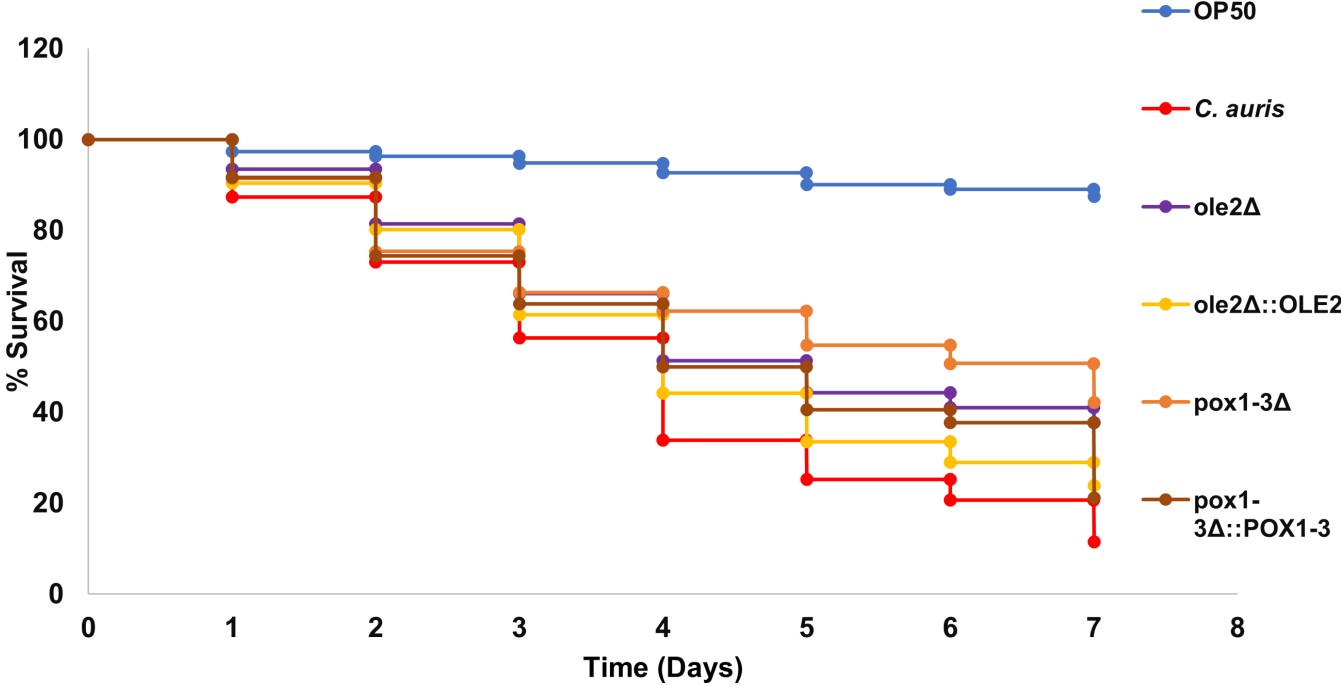

**FIG 4** Kaplan-Meier plot indicating the survival probability to assess the virulence of *C. auris* strains. *Escherichia coli* (OP50) was used as the control.

work may focus on absolute quantification of the various fatty acids to obtain clearer information regarding the role of Ole2.

Despite a similar effect on fatty acid profile between *C. parapsilosis* and *C. auris*, the effect of deletion of *OLE2* on $PGE_2$ synthesis in *C. auris* is different from *C. parapsilosis* and similar to that of *C. albicans*. The involvement of *OLE2* in the production of $PGE_2$ by *C. albicans* has been previously demonstrated as the disruption of the *OLE2* gene resulted in a marked reduction in $PGE_2$ production (11), but for *C. parapsilosis*, the deletion of the *OLE2* gene showed no decrease in $PGE_2$ production (16). In the case of *C. auris*, the deletion strain (*ole2Δ*) displayed a significantly reduced $PGE_2$ production compared to the wild-type and add-back mutant (*ole2Δ::OLE2*), with the latter restoring $PGE_2$ production to wild-type levels. This difference may be due to the lower percentage protein identity between Ole2 from *C. parapsilosis* and *C. auris* than between the proteins from *C. albicans* and *C. auris* (Table 3) or specific amino acid differences between the proteins.

POX1-3 encodes an acyl-CoA oxidase, which is also involved in the beta-oxidation of fatty acids by catalyzing the conversion of acyl-CoA to 2-trans-enoyl-CoA, hydrogen peroxide, and acetyl-CoA (28). Disruption of *POX1-3* significantly decreased $PGE_2$ production in *C. parapsilosis* (16, 28). This corresponds to our results since deletion of *POX1-3* in *C. auris* also caused reduced $PGE_2$ production compared to the wild type. The

**TABLE 2** Median lifespan, standard error, and time required to achieve 50% mortality in *C. auris*-infected nematodes

| Strain | Median lifespan (days) | Standard error | Days to reach 50% survival | Corrected Bonferroni *P*-value |
|---|---|---|---|---|
| *C. auris* MRU293 | 4.12 | 0.26 | 4 | –[b] |
| *ole2Δ* | 4.97 | 0.27 | 5 | 0.0168[a] |
| *ole2Δ::OLE2* | 4.53 | 0.27 | 4 | 0.9638 |
| *pox1-3Δ* | 5.15 | 0.3 | 7 | 0.0054[a] |
| *pox1-3Δ::POX1-3* | 4.68 | 0.29 | 5 | 0.8736 |

[a]Significant difference compared to *C. auris* MRU293.
[b]"–" indicate no comparison made.

**TABLE 3** Percentage identity of Ole2 and Pox1-3 from *C. albicans* and *C. parapsilosis* compared to *C. auris*

| Organism | Entry | Gene name | Length | Identity |
|---|---|---|---|---|
| *C. albicans* SC5314 | A0A1D8PHT9 | *OLE2*, C2_07090C_A | 526 aa | 56.6% |
| *C. parapsilosis* MYA-4646 | G8BJN8 | Cp*OLE2*, CPAR2_406570 | 550 aa | 55.1% |
| *C. albicans* SC5314 | Q5AJD9 | *POX1-3*, C3_01960C_A | 709 aa | 62.4% |
| *C. parapsilosis* MYA-4646 | G8BAZ7 | Cp*POX1-3*, CPAR2_807700 | 711 aa | 60.5% |

production was restored to wild-type levels in the add-back mutant (*pox1-3Δ::POX1-3*). In Table 3, it can also be seen that the percentage identity of Pox1-3 between *C. auris*, *C. albicans,* and *C. parapsilosis* is more than 60%, indicating that they are likely to perform similar functions in these three yeasts. Unfortunately, the increase in 16:0 and decrease in 18:1 observed for the *POX1-3* deletion mutant were not restored to wild-type levels in the add-back strain, indicating potential off-target effects of the deletion.

Interestingly, although the role of *OLE2* in lipid metabolism, including $PGE_2$ production, seems to differ between different yeast species, the role of this gene in virulence of different pathogenic yeasts across various infection models, including human monocyte-derived macrophages (16) and nematodes, is conserved, indicating that the role of this gene in full virulence may be independent of its role in $PGE_2$ production.

Deletion of *POX1-3* caused a decrease in $PGE_2$ production as well as virulence in both *C. parapsilosis* and *C. auris*. This implies that there is a link between the phenotypes, but it cannot be ruled out that it could be due to other metabolic consequences of this gene deletion. The addition of exogenous $PGE_2$ to this infection model may untangle these two phenotypes and provide clarity regarding the mechanisms by which these two genes influence virulence. However, since the role of $PGE_2$ in *C. elegans* and mammals is not necessarily conserved, with this eicosanoid playing a role in inflammation in mammals and in longevity in *C. elegans* (18), care should be taken when correlating such links in *C. elegans* to mammals. Nevertheless, these findings provide evidence of a conserved role of this gene in eicosanoid production, highlighting the conserved nature of $PGE_2$ production in pathogenic yeasts and laying the foundation for studying this unique metabolic pathway, which differs from mammalian hosts, in search of novel antifungal drug targets.

## AUTHOR AFFILIATION

[1]Department of Microbiology and Biochemistry, University of the Free State, Bloemfontein, South Africa

## AUTHOR ORCIDs

Carolina Pohl http://orcid.org/0000-0001-6928-5663
Jacobus Albertyn http://orcid.org/0000-0003-4334-5925

## FUNDING

| Funder | Grant(s) | Author(s) |
|---|---|---|
| National Research Foundation of South Africa | 115566 | Carolina Pohl |

## AUTHOR CONTRIBUTIONS

Armand Bolsenbroek, Investigation, Visualization, Writing – original draft | Eduvan Bisschoff, Investigation, Writing – original draft | Gabre Kemp, Formal analysis, Methodology | Olihile Sebolai, Supervision | Carolina Pohl, Conceptualization, Project administration, Resources, Supervision, Writing – review and editing | Jacobus Albertyn, Conceptualization, Methodology, Resources, Supervision, Writing – review and editing

## DATA AVAILABILITY

All data are available in the article and supplemental material.

## ADDITIONAL FILES

The following material is available online.

### Supplemental Material

**Supplemental material (Spectrum02323-25-s0001.docx).** Tables S1 to S4, Figures S1 to S12, and supplemental Materials and Methods.

### Open Peer Review

**PEER REVIEW HISTORY (review-history.pdf).** An accounting of the reviewer comments and feedback.

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
