## [Reviewer comments · Microbiology Spectrum]

Microbiology Spectrum

The role of *OLE2* and *POX1-3* in prostaglandin E₂ production and virulence is conserved in *Candidozyma (Candida) auris*

Armand Bolsenbroek, Eduvan Bisschoff, Gabre Kemp, Olihile Sebolai, Carolina Pohl, and Jakobus Albertyn

Corresponding Author(s): Carolina Pohl, University of the Free State

Review Timeline:

Submission Date:	July 29, 2025
Editorial Decision:	October 28, 2025
Revision Received:	December 4, 2025
Accepted:	December 19, 2025

Editor: Patricia Albuquerque

Reviewer(s): Disclosure of reviewer identity is with reference to reviewer comments included in decision letter(s). The following individuals involved in review of your submission have agreed to reveal their identity: Feng Yang (Reviewer #2)

Transaction Report:

DOI: <https://doi.org/10.1128/spectrum.02323-25>

Re: Spectrum02323-25 (**The role of *OLE2* and *POX1-3* in prostaglandin E₂ production and virulence is conserved in *Candidozyma (Candida) auris***)

Dear Prof. Carolina H Pohl:

Thank you for the privilege of reviewing your work. Below you will find my comments, instructions from the Spectrum editorial office, and the reviewer comments.

Please ensure that you address all of the reviewers' questions and comments in full, providing clear and detailed responses. In particular, the reviewers have emphasized the need for further characterization of the mutants and wild-type (WT), including:

Growth characteristics and comparative data

Relative lipid profile analysis

These additions are essential to strengthen the manuscript's conclusions and scientific rigor.

Additionally, please revise and improve the figures to meet publication-quality standards. Ensure that all figures are clear, properly labeled, and of high resolution suitable for print and online publication.

Revision Guidelines

Sincerely,
Patricia Albuquerque

Reviewer #1 (Comments for the Author):

C. albicans prostaglandin, PGE₂, is crucial for biofilm formation, colonization, and virulence despite biosynthetic pathway being unknown. OLE2 and POX1-3 have been implicated in PGE₂ production, and their role is investigated in the multi-drug-resistant fungal pathogen *Candidozyma* (*Candida*) *auris*, an escalating global health threat causing hospital outbreaks.

Deletion mutants were generated using CRISPR-Cas9 and PGE₂ production quantified using ELIZA and LC-MS. This is the first report of PGE₂ production in *Candidozyma*, and OLE2 and POX1-3 involvement in PGE₂ production and virulence in a *Caenorhabditis elegans* infection model. Combined with the findings of others, their results support roles for both genes in virulence that are most likely independent of their role in PGE₂ production.

Rationale for this study: PGE₂ is produced by *C. albicans*, *C. dubliniensis*, *C. parapsilosis*, *C. glabrata*, and *C. tropicalis*. However, OLE2 (encodes a putative stearyl coenzyme A desaturase) plays a role in PGE₂ production in *C. albicans* but not in *C. parapsilosis*. Hence the mechanism of PGE₂ production seems to differ among candida species and, in *Candidozyma*, is unknown.

Comments and questions

1. Figure 1: Font on the Y and X axis on the chromatograms are too small. It's unclear what the relevant transitions are? Figure 1 legend should specify that it is an MS chromatogram.
2. Figure 3 Font on X and Y axis, axis title and key are illegible. Figure quality is poor. This figure should not have passed QC. The last 2 FAs should be on a different scale. Figure 3 is difficult to evaluate without considerable scrutinization and guesswork as to which FAs they are referring to.
3. Since the decreases in PGE₂ production by the *Candidozyma* mutants are small, the exact P values for the t-test comparisons should be stated in Fig 2. Title for figure legend should be more informative i.e. quantification by ELIZA.
4. A good attempt was made to understand how the 2 genes are involved in PGE₂ production using FA profiling to investigate whether the gene products have desaturase activity. Like *C. parapsilosis*, it was found that deletion of OLE2 in *Candidozyma* leads to an accumulation of 16:1 and 18:1 at the expense of 16:0 and 18:0, providing further evidence that OLE2 does not encode a $\Delta 9$ desaturase. This information is important and should go in the abstract.
5. It would be good to include a sequence homology comparison table for OLE2 and POX1 in the different *Candida* species, particularly *albicans*, *parapsilosis* and *Candidozyma*.
6. Minor: Line 160-161 methods, Quantification of PGE₂ production by *Candidozyma*. Please specify where the collected supernatant came from

Reviewer #2 (Comments for the Author):

This is a well-executed and timely study that provides the first evidence of prostaglandin E₂ (PGE₂) production in the critical emerging fungal pathogen *Candida auris*. The authors effectively utilize a CRISPR-Cas9 system to demonstrate the roles of OLE2 and POX1-3 in PGE₂ production and virulence in a *Caenorhabditis elegans* infection model. The data are compelling, the methodology is sound, and the findings are significant for the field of fungal pathogenesis and the search for novel antifungal targets. The manuscript is generally clear and suitable for publication after addressing the points below.

Major Comments

The finding that the *ole2* Δ mutant accumulates monounsaturated fatty acids (MUFAs) is fascinating and counter-intuitive for a putative desaturase gene. While the authors correctly conclude that this indicates the gene does not encode a $\Delta 9$ desaturase, this intriguing phenotype warrants a more nuanced discussion. It would be valuable to speculate on the potential function of OLE2 in *C. auris*. Could it be involved in the downstream modification or turnover of MUFAs? Its role might be indirect, perhaps regulating the expression or activity of other enzymes in the fatty acid metabolic network. Expanding the discussion on this point would add significant depth.

The study elegantly shows that deletion of OLE2 or POX1-3 leads to both reduced PGE₂ production and attenuated virulence. However, the direct causal link between these two phenotypes is not fully established, and the authors appropriately note this for OLE2 (lines 291-295). To strengthen the manuscript, the discussion could explicitly acknowledge this limitation for both

genes. While the data strongly suggest a connection, it remains formally possible that the virulence defects are partly due to other, pleiotropic metabolic consequences of the gene deletions. A sentence framing the conclusions with this in mind would enhance the manuscript's rigor. Furthermore, could the authors comment on whether exogenous supplementation of PGE₂ in the infection model could help resolve this question in future studies?

Minor Comments

Ensure species names are presented in full the first time they appear, followed by the abbreviation in parentheses. Subsequently, the abbreviation can be used. Similarly, the abbreviation for arachidonic acid (AA) should be introduced at its first mention and used consistently thereafter, including in figure legends.

Line 36: Format as "Candidozyma auris (Candida auris)" for consistency and clarity.

Figure 2 Legend: Correct "SC5315" to "SC5314".

Reviewer #3 (Comments for the Author):

Here the authors have shown that the clinically significant yeast pathogen *C. auris* produces PGE₂ as has been previously reported for other human pathogenic yeast. Additionally, the authors demonstrate a role for both the desaturase OLE2 and the acyl-CoA oxidase POX1-3 in the production of this prostaglandin and in virulence. The deletion of either gene reduced the production of PGE₂ from exogenous arachidonic and altered the lipid profiles of this yeast.

Suggested Changes:

In the introduction, the authors discuss an impact of iron acquisition enzymes and FET3 on PGE₂ production by other *Candida* spp., but their work does not include any link to iron. The manuscript would benefit from a clearer justification for the selection of the genes OLE2 and POX1-3 over others mentioned.

Lines 134-147 suggest that biofilm and biomass experiments were performed for all strains. Were there significant differences in any of the mutants' growth *in vitro*?

A media + AA only control would be good to see at least for the data in Figure 1 and perhaps also Figure 3.

In line 219, I am not sure it can be said that "both OLE2 and POX1-3 are required" for the production of PGE₂ in *C. auris* given that the prostaglandin can still be detected in both knockout strains. Either the authors should generate a double KO mutant to assess if PGE₂ production is abolished entirely or they should rephrase the result to specify that they are required for "full" production.

The image quality of Figure 3 is very poor making it difficult to interpret.

It would be interesting to see how deletion of these genes impacted the absolute amounts of the lipids quantified in Figure 2. This could also be relevant to their roles in virulence.

In lines 237-239, the authors mention that the add-back POX1-3 strain did not restore WT lipid profiles; this should be addressed in the discussion.

The authors state in lines 212-213 that student t-tests were used for all non-survival statistical analyses, but there is no mention of correction for multiple testing. Should Bonferroni correction be considered for the data in Figures 2 and 3 where all strains were compared to the WT?

There is not a strong link between the role of OLE2 and POX1-3 in PGE₂ production and virulence. It would be interesting to see if supplementation of exogenous PGE₂ restores the virulence of either deletion mutant. The authors mention that this may suggest a PGE₂-independent role for OLE2 in *C. auris* virulence as previously published for *C. parapsilosis*. Discussion of a potential alternative mechanism based on the literature is also needed here, especially considering the difference in OLE2 function between the two species.

We would like to thank the reviewers for their insightful comments on the manuscript and the time they spent to read it. Please find our responses below:

Reviewer #1 (Comments for the Author):

1. Figure 1: Font on the Y and X axis on the chromatograms are too small. It's unclear what the relevant transitions are? Figure 1 legend should specify that it is an MS chromatogram.

Response: The font has been increased and the legend improved

2. Figure 3 Font on X and Y axis, axis title and key are illegible. Figure quality is poor. This figure should not have passed QC. The last 2 FAs should be on a different scale.

Figure 3 is difficult to evaluate without considerable scrutinization and guesswork as to which FAs they are referring to.

Response: This figure has been improved and the last two fatty acids provided with their own y-axis as requested

3. Since the decreases in PGE2 production by the *Candidozyma* mutants are small, the exact P values for the t-test comparisons should be stated in Fig 2. Title for figure legend should be more informative i.e. quantification by ELIZA.

Response: The legend has been improved as suggested. The specific P values have also been indicated in the legend

4. A good attempt was made to understand how the 2 genes are involved in PGE2 production using FA profiling to investigate whether the gene products have desaturase activity. Like *C. parapsilosis*, it was found that deletion of OLE2 in *Candidozyma*

leads to an accumulation of 16:1 and 18:1 at the expense of 16:0 and 18:0, providing further evidence that OLE2 does not encode a $\Delta 9$ desaturase. This information is important and should go in the abstract.

Response: This information has been added to the abstract as requested

5. It would be good to include a sequence homology comparison table for OLE2 and POX1 in the different *Candida* species, particularly *albicans*, *parapsilosis* and *Candidozyma*.

Response: A table (Table 3) has been included providing the % protein identity of the genes from *C. albicans* and *C. parapsilosis* compared to those of *C. auris*. This information has also been included in the discussion

6. Minor: Line 160-161 methods, Quantification of PGE₂ production by *Candidozyma*.

Please specify where the collected supernatant came from

Response: Corrected to indicate the source of the supernatants

Reviewer #2 (Comments for the Author):

Major Comments

1. The finding that the *ole2Δ* mutant accumulates monounsaturated fatty acids (MUFAs) is fascinating and counter-intuitive for a putative desaturase gene. While the authors correctly conclude that this indicates the gene does not encode a $\Delta 9$ desaturase, this intriguing phenotype warrants a more nuanced discussion. It would be valuable to speculate on the potential function of OLE2 in *C. auris*. Could it be involved in the downstream modification or turnover of MUFAs? Its role might be indirect, perhaps regulating the expression or activity of other enzymes in the fatty acid metabolic network. Expanding the discussion on this point would add significant depth.

Response: The discussion has been expanded as requested

2. The study elegantly shows that deletion of OLE2 or POX1-3 leads to both reduced PGE₂ production and attenuated virulence. However, the direct causal link between these two phenotypes is not fully established, and the authors appropriately note this for OLE2 (lines 291-295). To strengthen the manuscript, the discussion could explicitly acknowledge this limitation for both genes. While the data strongly suggest a connection, it remains formally possible that the virulence defects are partly due to other, pleiotropic metabolic consequences of the gene deletions. A sentence framing the conclusions with this in mind would enhance the manuscript's rigor. Furthermore, could the authors comment on whether exogenous supplementation of PGE₂ in the infection model could help resolve this question in future studies?

Response: The discussion has been amended as suggested.

Minor Comments

3. Ensure species names are presented in full the first time they appear, followed by the abbreviation in parentheses. Subsequently, the abbreviation can be used. Similarly, the abbreviation for arachidonic acid (AA) should be introduced at its first mention and used consistently thereafter, including in figure legends.

Response: This has been revised as requested

Line 36: Format as "Candidozyma auris (Candida auris)" for consistency and clarity.

Response: This has been revised as requested

Figure 2 Legend: Correct "SC5315" to "SC5314".

Response: This has been revised as requested

Reviewer #3 (Comments for the Author)

Suggested Changes:

1. In the introduction, the authors discuss an impact of iron acquisition enzymes and FET3 on PGE₂ production by other *Candida* spp., but their work does not include any link to iron. The manuscript would benefit from a clearer justification for the selection of the genes OLE2 and POX1-3 over others mentioned.

Response: The mention of the iron acquisition enzymes has been modified. The reason they are included is to show the similarity of some genes (*FETs*) involved in PGE₂ synthesis in different species. The choice of the genes in this study has been better motivated

2. Lines 134-147 suggest that biofilm and biomass experiments were performed for all strains. Were there significant differences in any of the mutants' growth *in vitro*?

Response: Although *ole2Δ* did show lower biofilm biomass production, this was not statistically significant. This data is presented in Supplementary figure S 12 as also now indicated in the manuscript

3. A media + AA only control would be good to see at least for the data in Figure 1 and perhaps also Figure 3.

Response: Figure 1 serves to indicate that the eicosanoid produced by *C. auris* is authentic PGE₂ as shown by the same transitions as present in the standard. The media + AA control was incorporated into the ELISA and these background values subtracted from the experimental values – this has been indicated in the relevant materials and methods section. Figure 3 is the fatty acid profiles of the strains without the addition of AA. The legend has been corrected to reflect this

4. In line 219, I am not sure it can be said that "both OLE2 and POX1-3 are required" for the production of PGE₂ in *C. auris* given that the prostaglandin can still be detected in both knockout strains. Either the authors should generate a double KO

mutant to assess if PGE₂ production is abolished entirely or they should rephrase the result to specify that they are required for "full" production.

Response: The heading has been changed as well as the phrasing in the paragraph, to better reflect that these either of these genes are required for full production

5. The image quality of Figure 3 is very poor making it difficult to interpret.

Response: The quality of figure 3 has been improved as also requested by reviewer 1

6. It would be interesting to see how deletion of these genes impacted the absolute amounts of the lipids quantified in Figure 2. This could also be relevant to their roles in virulence.

Response: This has been added as potential future work

6. In lines 237-239, the authors mention that the add-back POX1-3 strain did not restore WT lipid profiles; this should be addressed in the discussion.

Response: This has been added as requested

7. The authors state in lines 212-213 that student t-tests were used for all non-survival statistical analyses, but there is no mention of correction for multiple testing. Should Bonferroni correction be considered for the data in Figures 2 and 3 where all strains were compared to the WT?

Response: The data were re-analysed using the Bonferroni correction.

7. There is not a strong link between the role of OLE2 and POX1-3 in PGE₂ production and virulence. It would be interesting to see if supplementation of exogenous PGE₂ restores the virulence of either deletion mutant. The authors mention that this may suggest a PGE₂-independent role for OLE2 in *C. auris* virulence as previously published for *C. parapsilosis*. Discussion of a potential alternative mechanism based on the literature is also needed here, especially considering the difference in OLE2 function between the two species.

Response: The possible addition of exogenous PGE₂ has been addressed in the discussion. Some speculation regarding the role of Ole2 in the activity of Fad3 was added, however unfortunately, too little is known about Ole2 in *Candida* species to be able to speculate further on possible alternative mechanisms.

Re: Spectrum02323-25R1 (**The role of *OLE2* and *POX1-3* in prostaglandin E₂ production and virulence is conserved in *Candidozyma (Candida) auris***)

Dear Prof. Carolina H Pohl:

Your manuscript has been accepted, and I am forwarding it to the ASM production staff for publication. Your paper will first be checked to make sure all elements meet the technical requirements. ASM staff will contact you if anything needs to be revised before copyediting and production can begin. Otherwise, you will be notified when your proofs are ready to be viewed.

Sincerely,
Patricia Albuquerque
Editor
Microbiology Spectrum

Reviewer #1 (Comments for the Author):

The study is technically sound and has been performed with experimental and methodological rigor

Reviewer #2 (Comments for the Author):

All my concerns have been addressed.